materials science/nanotechnology/electron microscopy

bacterial cellulose, multi-walled carbon nanotube/zinc oxide, composite additive, hybrid membrane, controllable properties

**Author for correspondence:**
Zoltán Németh
e-mail: kemnemet@uni-miskolc.hu

This article has been edited by the Royal Society of Chemistry, including the commissioning, peer review process and editorial aspects up to the point of acceptance.

[†]Present address: Institute of Chemistry, University of Miskolc, Egyetemváros út 1. Miskolc-Egyetemváros, H-3515, Hungary.

# Development of bacterial cellulose–ZnO–MWCNT hybrid membranes: a study of structural and mechanical properties

Bilal El Mrabate[1,2], Mahitha Udayakumar[1,2], Emília Csiszár[5], Ferenc Kristály[3], Máté Leskó[3], László Somlyai Sipos[4], Mateusz Schabikowski[6] and Zoltán Németh[1,2,†]

[1]Higher Education and Industry Cooperation Center of Advanced Materials and Intelligent Technologies, [2]Institute of Chemistry, [3]Institute of Mineralogy and Geology, and [4]Institute of Physical Metallurgy, Metal Forming and Nanotechnology, University of Miskolc, H-3515 Miskolc, Hungary
[5]Department of Physical Chemistry and Materials Science, Budapest University of Technology and Economics, Műegyetem rkp. 3, H-1111 Budapest, Hungary
[6]Institute of Nuclear Physics, Polish Academy of Sciences, PL 31-342 Krakow, Poland

FK, 0000-0002-0075-5994; ZN, 0000-0001-7871-3818

Self-supported and flexible bacterial cellulose (BC) based hybrid membranes were synthesized and decorated with zinc oxide/multi-walled carbon nanotube (ZnO–MWCNT) composite additives in order to modify and tune their surface and bulk properties. Two types of ZnO–MWCNT additives with different morphologies were used in a wide concentration range from 0 to 90% for BC-based hybrids produced by filtration. The interaction between BC and ZnO–MWCNT and the effect of concentration and morphology of additives on the properties like zeta potential, hydrophilicity, electrical conductivity, etc. would be an important factor in various applications. Furthermore, the as-prepared hybrid membranes were characterized with the use of scanning electron microscopy (SEM), focused ion beam scanning electron microscopy (FIB-SEM), energy dispersive X-ray spectroscopy (EDS), X-ray powder diffraction (XRD) and surface area measurement (BET). Applying the presented synthesis routes, the surface properties of BC-based membranes can be tailored easily. Results reveal that the as-prepared BC–ZnO–MWCNT hybrid membranes can be ideal candidates for different kinds of applications, such as water filtration or catalysts.

# 1. Introduction

Over the last decades, the increased contribution of membrane separation technologies opened new pathways to wastewater treatment [1]. Gravity-driven membrane processes have been proved to be the vanguard of water purification methods [2]. To overcome many of the critical limitations of traditional membranes, such as selectivity, regeneration and recycling, there is a continuous search for a new type of membrane materials by increasing the involvement of nanotechnology [3]. In this context, the potential of the most abundant biopolymer available on Earth, cellulose, has been continuously increasing.

Cellulosic materials have many advantages: they are environmentally friendly and renewable; they can be considered as sustainable and cost-effective sources of composites and support material for the development of hybrid membrane materials [4]. Recently, cellulose-based nanomaterials, such as bacterial cellulose (BC), cellulose nanocrystals (CNC) and cellulose nanofibres (CNF), as well as their modification and use in value-added applications have attracted significant attention of both academy and the industry.

Nanofibres of cellulose produced by different bacteria can be considered as pure cellulose, with high crystallinity, unique morphology, uniform diameter, excellent tensile strength and high modulus [5]. These outstanding properties make the BC nanofibres possible candidates to develop high-performance composites and membranes for various application areas [6]. A recent study reported different methods to produce BC nanocomposites with the emphasis on the processing techniques that allow the incorporation of functional nanofillers and additional phases without disturbing the original BC network structure [7]. To increase the properties of cellulose-based membranes and to modify their performance and selectivity in chemical and physical processes, decoration of cellulose derived from various sources with inorganic nanoparticles such as zinc oxide (ZnO) [8,9] or multi-walled carbon nanotube (MWCNT) [10], has become an important aspect of the research.

BC-based nanocomposites were prepared with metal oxides such as $TiO_2$, ZnO and CuO for developing antibacterial and photocatalytic activity [11–14]. A novel study presented the synergistic effect of ZnO nanoparticles and propolis extracts deposited on BC [15]. $Fe_2O_3$–$SnO_2$/BC nanohybrid composite was also synthesized by incorporating the iron–tin binary oxide into a cellulosic framework [16]. Moreover, biosynthesized spherical $Fe_3O_4$/BC nanocomposites were applied as adsorbents for the removal of heavy metal ions [17].

In addition, previously published papers described the properties and some possible applications of membranes produced by combining cellulose and MWCNT. Since MWCNT has also exceptional properties (e.g. high surface area, thermal stability, outstanding adsorption capacity, thermal and electrical conductivity), its use is increasing continuously in the field of composite membrane technology [18]. The surface of regenerated cellulose fibres (namely commercial viscose fibres) with a micrometre-sized diameter and cellulose II crystal structure was modified with MWCNT by a simple dip coating process [10] and the sensing behaviour of fibres to volatile molecules and their applicability in the area of smart textiles and wearable technology were investigated. It was also presented that the incorporation of MWCNTs into fibrous cellulose network is a useful method to enhance the electrical conductivity of the cellulose membranes [19], although the usage of blank MWCNT in composite membrane development has a great challenge, especially due to the high production cost and poor wetting properties.

Based on the results mentioned above, it can be assumed that combining the remarkable properties both of ZnO particles (e.g. photoactivity, low toxicity to human body, etc.) and MWCNT (e.g. high surface area, conductivity) by preparation of ZnO–MWCNT nanocomposites and incorporating them into the cellulose matrix can provide new directions in the development of hybrid membranes. By the ZnO–MWCNT composite additives, the surface properties of membranes can be controlled and modified easily. However, little is known about the special properties and possible application areas of such cellulose-based hybrid membranes.

Thus, the aim of this work was to synthetize BC-based hybrid membranes with ZnO–MWCNT nanocomposite additives with special surface properties and various functionalities. ZnO–MWCNT composite additives of different morphology were used depending upon the preparation methods (e.g. impregnation and solvothermal). Results reveal that BC nanofibres with nano-sized diameter and cellulose I crystal structure acted as an ideal framework for holding the ZnO–MWCNT composite and contributed to creation of self-supported membranes. By applying the ZnO–MWCNT additives in a wide concentration range, surface and bulk properties of BC can be modified and tuned to achieve the desired properties to hybrid membranes.

# 2. Material and methods

## 2.1. Materials

Commercially available MWCNT (NC7000), with a diameter of 30–120 nm, the length of 1.5–2.5 µm and the surface area of 300 $m^2 g^{-1}$ was purchased from Nanocyl SA, Belgium. The precursor compound, zinc acetate [$Zn(CH_3COO)_2 \times 2H_2O$] (ZnAc), the solvent, absolute ethanol (EtOH), the surfactant, cetyltrimethylammonium bromide (CTAB) were used without further purification. All of these chemicals were purchased from VWR Chemicals, Hungary. HCl of 2 M and MilliQ water (18.2 MΩ cm) were used through the experiments. BC fibres were received from nata de coco cubes (FI, Philippines) by an alkaline and oxidative purification and a mechanical disintegration [20]. To prepare the membranes, an aqueous suspension of BC with a dry solid content of 1% was used. Polytetrafluoroethylene (PTFE) filter of the pore size of 0.22 µm and the diameter of 47 mm (Durapore-GVWP04700) was used for membrane preparation.

## 2.2. Preparation of ZnO–MWCNT composite additives

Two different synthesis pathways—impregnation and solvothermal methods—were applied to prepare the additives, which were selected on the basis of our recent results [21,22]. Briefly, the exact calculated amount of purified MWCNT (100 mg) was sonicated in 100 ml of EtOH using an ultrasonic laboratory homogenizer (40 kHz and output power of 40 W; Heilsher GmbH, Germany) for 15 min. The calculated amount of zinc acetate (2.42 g) was dissolved in 50 ml of EtOH and left under vigorous stirring of a magnetic stirrer for 15 min at 300 r.p.m. to ensure complete dissolution. The MWCNT content in the total mass of the final composites was set at 10 w/w%.

For impregnation synthesis, the solution of precursor was added dropwise (at 1 ml $min^{-1}$) to the suspension of MWCNT. Then, the mixture was allowed to react under mechanical stirring for 12 h; afterwards, it was heated up to 75°C to remove the solvents. Eventually, the final product was calcinated at 400°C for 4 h in a static furnace, at 5°C $min^{-1}$ heating rate, to obtain ZnO nanoparticles on the surface of MWCNT.

For solvothermal synthesis, the MWCNT suspension was added directly to the precursor solution under stirring. Then the mixture was poured into 150 ml stainless steel autoclaves with an attached Teflon tube and placed in a static furnace for 12 h at 150°C. Subsequently, the final product was centrifuged and washed to remove residues and calcinated for 4 h at 400°C at 5°C $min^{-1}$ heating rate. The final products prepared by impregnation and solvothermal were denoted ZnO–MWCNT–IMP and ZnO–MWCNT–SOLVO, respectively.

The ZnO–MWCNT composite additives prepared by both techniques were treated with CTAB in order to modify their surface properties and to stabilize them against van der Waals attraction [23,24]. The treatment was performed by adding 0.1 g of ZnO–MWCNT composite material to 50 ml of the CTAB solution (0.05 M) in distilled water. The obtained mixture was sonicated for 10 min and subsequently was stirred for 4 h. Then, it was centrifuged and washed with deionized water to achieve neutral pH. Finally, the obtained powders were dried in an oven at 100°C for 12 h.

## 2.3. Preparation of BC–ZnO–MWCNT hybrid membranes

To prepare the BC–ZnO–MWCNT nanohybrid membranes, the calculated amount of BC fibres was dipped into 15 ml EtOH. Meanwhile, the ZnO–MWCNT–IMP and ZnO–MWCNT–SOLVO composite additives were sonicated in 50 ml EtOH for 15 min. The content of the composite additives in the final nanohybrid membrane was varied in a wide range from 10 to 90 w/w%. Then the BC suspension was added to the ZnO–MWCNT–IMP and the ZnO–MWCNT–SOLVO suspensions and the mixture was mechanically stirred for 5 h at 300 r.p.m. Finally, the preparation of BC-based hybrid membranes was accomplished by vacuum filtration through a PTFE membrane to achieve a loading of 8 mg $cm^{-2}$ (total mass 100 mg/membrane), then it was dried in air. Hybrid membranes were denoted as IMP 10–90 and SOLVO 10–90 depending on the preparation techniques (IMP, SOLVO) and the amount of applied composite additive (10–90%).

## 2.4. Characterization of membranes

The morphology of hybrid membranes was verified by scanning electron microscopy (SEM) and focus ion beam scanning electron microscopy (FIB-SEM). SEM investigation was performed with the use of

Hitachi S-4800 Type II FE-SEM operating in the range of 0–10 keV. Energy dispersive X-ray spectroscopy (EDS) measurement was completed by a scanning electron microscope and a Röntec XFlash Detector 3001 SDD device. FIB-SEM measurements were done with a Thermo Helios G4 PFIB CXe instrument.

Crystalline structure was determined by X-ray powder diffraction (XRD) method (Bruker D8 Advance diffractometer) with Cu Kα radiation (40 kV and 40 mA) in parallel beam geometry (Göbel mirror) with position sensitive detector (Vantec1, 1° opening). Measurements were taken in the 2–100° ($2\theta$) range with 0.007° ($2\theta$)/14 s goniometer speed, on top-loaded specimens in zero-background Si sample holders. Raman microscopy measurements were performed by a high-resolution Raman spectrometer Nicolet Almega XR (Thermo Electron Corporation, USA) with a 532 nm Nd:YAG laser (50 mW). Nitrogen ($N_2$) adsorption–desorption experiment was performed at 77 K to determine the surface area and micropore volume (t-plot method) using (ASAP 2020, Micromeritics Instrument Corp. Germany). Prior to each measurement, the samples were degassed at 90°C for 24 h.

Electrophoretic measurement was performed by dynamic light scattering (DLS) (ZetaSizer NS, Malvern, UK) device. The measurement is based on the combination of laser Doppler velocimetry and phase analysis of light scattering (PALS) in Malvern's M3-PALS technique. Contact angle measurement was carried out by using the sessile drop method (SP 12 melt microscope, Sunplant Ltd, Hungary), making a silhouette shot. KSV software (KSV Instrument Ltd, Finland) was used to evaluate the recording and determine the angular values.

Colour of the composite membranes was measured by using a HunterLab Color Quest XE (USA) colorimeter (D65/10°) and compared to the white background tile of apparatus. Colour difference ($\Delta E_{ab}{}^*$) and lightness ($L^*$) values were determined in CIELab colour space. Water vapour sorption is usually used for characterization of the internal surfaces of cellulose-based substrates available for water vapour molecules and determined by exposing the samples, previously dried over $P_2O_5$ for 5 days, to an atmosphere of 65% RH at 25°C for 5 days and calculated as percentage of the dry weight.

Mechanical properties were investigated by an Instron 5566 tensile tester (USA) at 10 mm min$^{-1}$ cross-head speed and 10 mm gauge length on specimens with 5 mm width. At least four parallel determinations were made and averaged. Electrical conductivity measurements were performed by a LabView-based measurement system. The measurement set-up consists of an NI PXI-1044 chassis, an NI PXI-8106 embedded controller module and an NI PXI-4071 high-precision DMM (digital multimeter) module. To measure the resistance (conductance) of the sample the Kelvin (four-wire) method is used.

# 3. Results and discussion

## 3.1. SEM, FIB-SEM and EDS analysis

ZnO–MWCNT additives prepared either by impregnation or by solvothermal methods were characterized carefully in a previous study [22]. It was stated that both ZnO nanoparticles (average 20–30 nm) and bigger hexagonal ZnO crystals (approx. 4–5 µm) evolved among the carbon nanotubes by using the above-mentioned synthesis routes.

Scanning electron micrographs in figure 1 show the surface structure of neat BC membrane (*a*) and hybrid membranes (*b*,*c*,*e*,*f*). The fabrication of membranes was successful in all cases, although different morphology was observed during SEM investigations. The as-prepared hybrid membranes were stable, flexible and self-supported, except for the membranes with the highest additive content (IMP 90 and SOLVO 90). These properties can be explained by the interaction between the certain amount of hydroxyl (–OH) groups on BC fibres and the oxygen-containing species on the surface of MWCNT, which resulted in interfacial bonding [10,19].

SEM images in figure 1*b*,*c* demonstrate the presence of ZnO nanoparticles on the surface of BC-based membranes prepared by impregnation method. In some cases, the ZnO particles stacked together resulting in bigger ZnO agglomerates. Applying ZnO–MWCNT–SOLVO additives (figure 1*e*,*f*) the produced hybrid membrane consists of regular hexagonal ZnO particles with an average size of 4 µm, and presumably the MWCNT was built into ZnO crystals.

To characterize the presence of ZnO particles on the surface of MWCNT, EDS analysis was performed for each sample. EDS spectra in figure 1*d* and *g* reveal that the detected signals originated from carbon (C), oxygen (O) and zinc (Zn), confirming the presence of BC, MWCNT and ZnO in the hybrid membranes. Furthermore, in order to demonstrate the differences between the distribution and particle size of ZnO crystals elemental mapping analysis (figure 1*d*,*g*) were recorded for atoms of

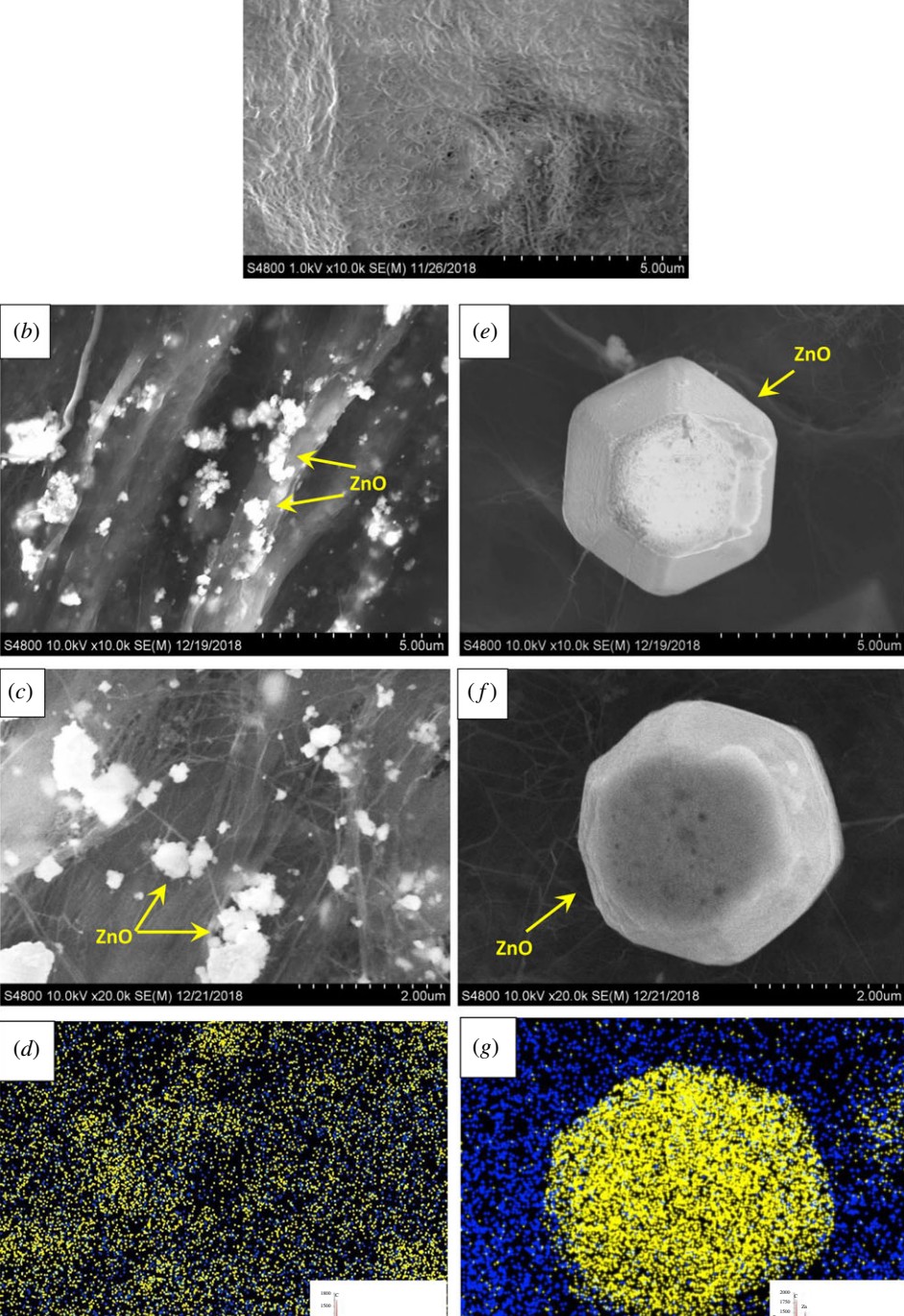

**Figure 1.** SEM micrographs of neat BC membrane (*a*) and BC–ZnO–MWCNT hybrid membranes: IMP 20 (*b,c*) and SOLVO 20 (*e,f*). EDS spectra and elemental mapping of hybrid membranes: IMP 20 (*d*) and SOLVO 20 (*g*).

interest: C as representative both of MWCNT and BC, as well as Zn for ZnO. It was observed that the concentration and distribution of Zn showed significant difference, thus confirming the different surface morphology of hybrid membranes. To answer the question whether the MWCNT is built into the inner structure of the hexagonal ZnO particles, thereby influencing the properties of BC-based hybrid membranes, FIB-SEM was applied for the SOLVO 20 nanohybrid membrane. Figure 2*a*–*c* shows the representative SEM images of the outer (figure 2*a*) and internal (figure 2*b,c*) structure of

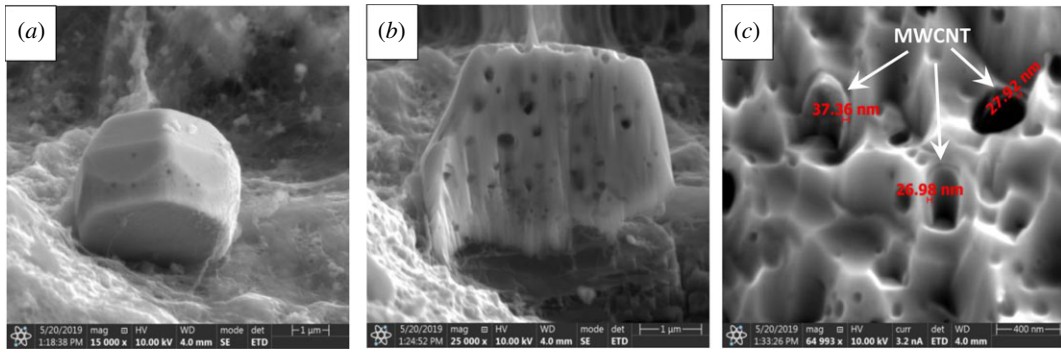

**Figure 2.** FIB-SEM micrographs of SOLVO 20 hybrid membrane: ZnO microparticle (*a*), cut ZnO particle (*b*), internal structure of ZnO. MWCNTs with an average diameter of 30 nm are marked with white arrows (*c*).

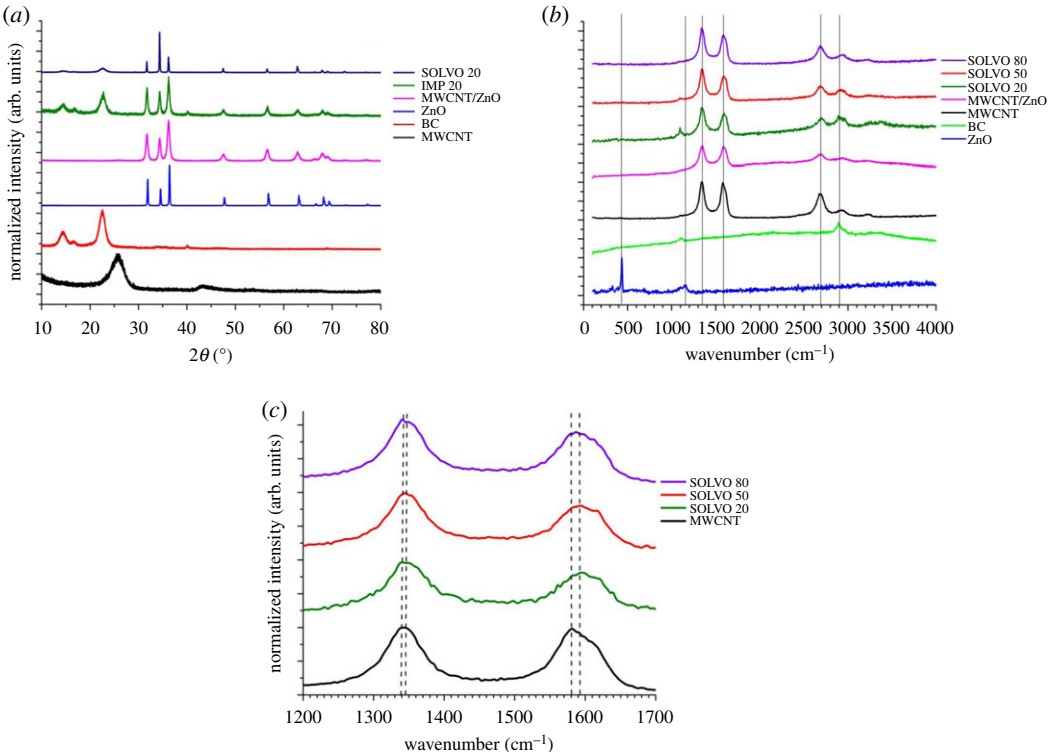

**Figure 3.** XRD (*a*) and Raman spectra (*b*,*c*) of raw materials (BC, ZnO and MWCNT), ZnO–MWCNT additives and BC-based hybrid membranes.

ZnO particles. The ZnO hexagonal particle was previously cut to gain information about their morphology. As can be seen in figure 2*b* the internal structure of the crystals contains numerous holes. Presumably, this phenomenon could be related to the presence of MWCNT and the crystallization process of ZnO particles. These cavities were filled with a tubular material with a wall thickness of approximately 30 nm as it was observed on the cross-section analysis in figure 2*c*. Measured values show good correlation with the average diameter of the MWCNT, confirming the presence of carbon nanotubes in the inner structure of the hexagonal ZnO. Furthermore, the results of EDS mapping analysis also confirm the presence of MWCNT inside the ZnO particles as the carbon signals were recorded in the ZnO particle (figure 1*g*).

## 3.2. X-ray and Raman spectroscopy and surface area measurement

In order to transform the amorphous phase of ZnO into crystalline, the composite additives were heat treated at 400°C for 4 h before using for decoration of BC membrane. The diffraction peaks at 2*θ* of 26.5° and 44° can be attributed to the graphite structure of MWCNT (figure 3*a*, black curve). The

**Table 1.** Summary of the ratios of D, G and G′ peaks.

|  | MWCNT | SOLVO 20 | SOLVO 50 | SOLVO 80 |
|---|---|---|---|---|
| $I_D/I_G$ | 1.02 | 1.04 | 1.05 | 1.04 |
| $I_{G'}/I_G$ | 0.89 | 0.97 | 0.96 | 0.96 |
| $I_D/I_{G'}$ | 1.14 | 1.07 | 1.09 | 1.08 |

characteristic peak of BC (figure 3a, red curve) existed at 2θ of 14.5°, 16.4° and 22.5°, corresponding to the diffraction plane (101) (amorphous region), (10$\bar{1}$) (amorphous region) and (200) (crystalline region), respectively [23–25]. According to Crystallography Open Database No. 9004178, diffraction peaks at 2θ of 31.7°, 34.5°, 36.1°, 47.5°, 56.7°, 62.8° and 67.9° correspond to the (100), (002), (101), (102), (110), (103) and (200) planes, respectively, reflections of hexagonal ZnO (figure 3a, blue curve) [26]. They are present in the diffractograms of ZnO-containing additive (figure 3a, purple curve) as well as in the IMP 20 and SOLVO 20 hybrid membranes (figure 3a, green and deep blue curves, respectively).

Particle size was calculated from the XRD results, using Rietveld refinement in Bruker TOPAS4, given as weight mean column length. The average particle diameter of ZnO crystallites is 20 ± 5 nm in the IMP hybrid membranes, whereas it is 260 ± 58 nm when solvothermal method was applied during preparation. Based on the particle size analysis and the SEM study (figure 1e,f) presented above, it can be concluded that ZnO crystallites form larger (4–5 µm) particles of hexagonal structure in SOLVO hybrid membranes than in membranes prepared by impregnation method [22]. The lattice parameters of ZnO were determined during Rietveld refinement, values for the impregnated samples are on average $a = 3.251 ± 0.001$ Å and $b = 5.210 ± 0.002$ Å, while the solvothermal synthesis shows $a = 3.250 ± 0.001$ Å and $b = 5.207 ± 0.001$ Å. The theoretical lattice parameters (COD 9004178) are $a = 3.249$ Å and $b = 5.203$ Å, in good agreement with the refined values.

In order to prove the chemical interaction between BC and ZnO–MWCNT composite additives, Raman spectra of the pristine materials (BC, MWCNT and ZnO) and hybrid membranes were recorded (figure 3b,c). The characteristic peaks of ZnO can be identified at 438 and 1156 cm$^{-1}$ (figure 3b, blue curve). The green curve at figure 3b shows peaks at 1105 and 2897 cm$^{-1}$ which are the characteristic bands of BC. Increasing the amount of additives, the intensity of the two bands of BC decreased considerably and the peaks became almost invisible in the case of hybrid membranes with high ZnO–MWCNT content. Three other dominant peaks deriving from MWCNTs (figure 3b, black curve) appear at 1339, 1580 and 2683 cm$^{-1}$. They are attributed to the D-, G- and G′-bands of MWCNTs, respectively. By calculating the ratio of these bands, the purity of the MWCNT samples can be easily determined [27]. As can be seen in figure 3c and table 1 significant changes were not detected by comparison of the intensity ratios of D/G, G′/G and D/G′, indicating that the structure of the composite additive was not damaged during the membrane preparation procedure [28]. The D-band is assigned to disorder induced by defects and curvature in the nanotube lattice, the G-band is the in-plane vibration of the C–C bonds, while the G′ band is the overtone of D-band [29]. Moreover, in figure 3c, up-shift in both D-band (from 1339 to 1345 cm$^{-1}$) and G-band (from 1580 to 1592 cm$^{-1}$) was observed in the spectra of hybrid membranes. This phenomenon could be explained by the formation of non-covalent interactions, presumably hydrogen bonds, between the MWCNT and BC fibres, resulting in a light band shifting [28].

As the results in table 2 reveal, the specific surface area of BC membranes dried from water-swollen state was found to be about 0.8 m$^2$ g$^{-1}$. A similar value was obtained for cotton (1 m$^2$ g$^{-1}$) dried also from water-swollen state [30] and a slightly higher value (2.7 m$^2$ g$^{-1}$) was measured for BC [31]. The commercial MWCNT has very high specific surface area (302 m$^2$ g$^{-1}$) determined also by nitrogen sorption. For composite additives, however, where the MWCNT is incorporated with ZnO, the values are significantly lower and depend largely on the preparation methods (IMP: 27 m$^2$ g$^{-1}$; SOLVO: 48 m$^2$ g$^{-1}$). Thus, the surface area of hybrid membranes is higher than that of the BC and lower than that of the composite additives.

## 3.3. Zeta potential and contact angle measurement

Zeta potential (ζ) of the hybrid membranes and raw materials was measured in the pH range of 3.0–9.0 and the results are presented in figure 4. Prior to the experiments, the samples were dispersed in

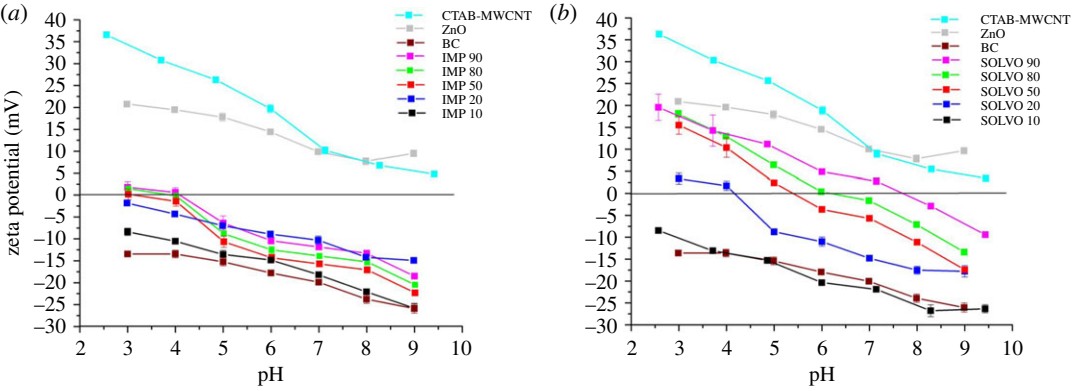

**Figure 4.** Zeta potential of raw materials: IMP (*a*) and SOLVO (*b*) hybrid membranes as a function of pH.

**Table 2.** Specific surface area determined by nitrogen sorption and isoelectric points (IEP) of the raw materials, composite additives and BC-based hybrid membranes.

| sample | specific surface area (m$^2$ g$^{-1}$) | IEP (pH) |
|---|---|---|
| MWCNT | 302 | 4.2 |
| CTAB-MWCNT | — | >9 |
| ZnO | 50 | >9 |
| BC | 1 | <3 |
| IMP 10 | 3 | <3 |
| IMP 20 | 6 | <3 |
| IMP 50 | 13 | 3.2 |
| IMP 80 | 21 | 4.0 |
| IMP 90 | 25 | 4.2 |
| SOLVO 10 | 13 | <3 |
| SOLVO 20 | 17 | 4.4 |
| SOLVO 50 | 25 | 5.5 |
| SOLVO 80 | 33 | 6.2 |
| SOLVO 90 | 36 | 7.5 |

deionized water to reach the final concentration of 0.1 wt%. The pH was adjusted by the addition of either HCl or NaOH solution (0.1 M). Each measurement was repeated three times. The zeta potential of the neat BC is negative whereas the surface-treated MWCNT and ZnO are positively charged in the selected range of pH. Decoration of BC with ZnO–MWCNT composite additives shifts their isoelectric point (IEP) towards higher values (the surface becomes more positively charged) which can be beneficial for example in water purification [32].

Comparing the results for each pH it can be concluded that the $\zeta$ of hybrid membranes of both series is increasing continuously with increasing the amount of the ZnO–MWCNT composite additive. It can also be ascertained that the SOLVO hybrid membranes have significantly higher IEP values (table 2) than the IMP hybrid membranes in the selected pH range and at the same ZnO–MWCNT additive content.

Surface hydrophilicity/hydrophobicity of the membranes was also characterized by water contact angles. Results reveal that the contact angles of IMP membranes were in the range of 36–42° (figure 5*a–c*), the surface is hydrophilic and the effect of additive content is negligible. It is obvious that SOLVO membranes exhibited higher water contact angles than IMP membranes. The contact angle changed from 77° to 136° (figure 5*d–f*) depending on the membrane composition. Consequently, the hydrophilic surface of the as-prepared hybrid membranes can be easily influenced by incorporating different amount of ZnO–MWCNT additive prepared by solvothermal method into the BC network.

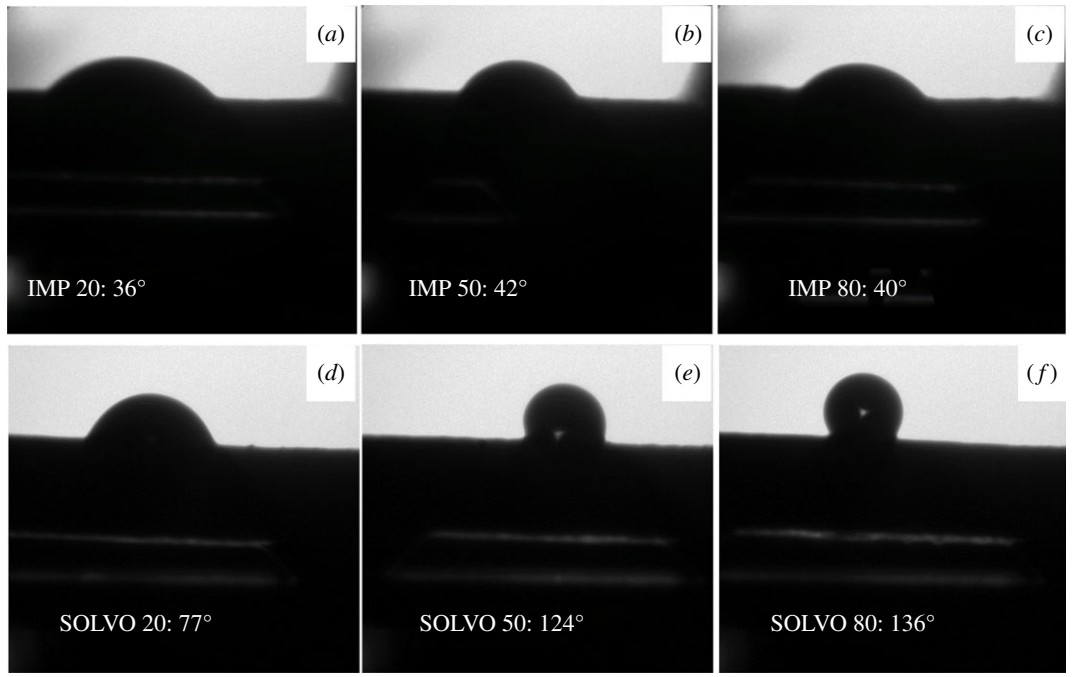

**Figure 5.** Photomicrographs of water drop with contact angle values measured on the surface of IMP (*a–c*) and SOLVO (*d–f*) hybrid membranes.

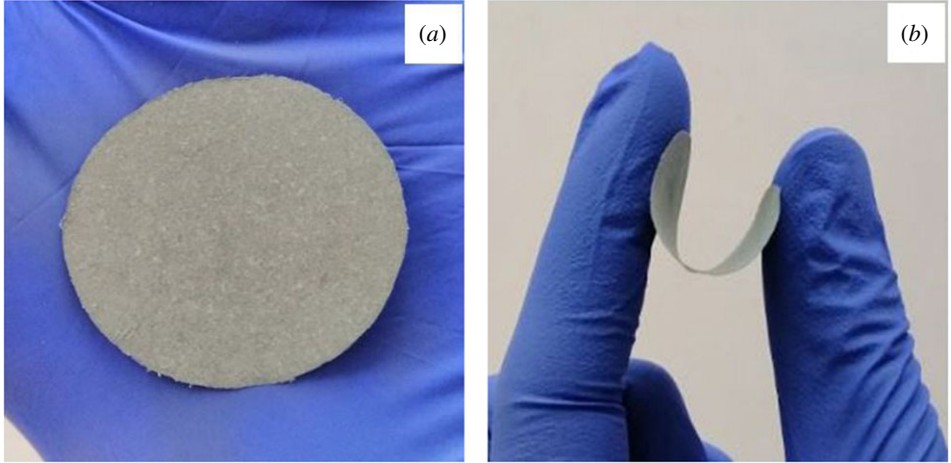

**Figure 6.** Representative photographs of SOLVO 20 hybrid membranes presenting the colour (*a*) and flexibility (*b*).

## 3.4. Colour and water vapour absorption capacity measurements

Smooth, flexible and grey membranes with a thickness of about 40–60 µm are presented in figure 6. It is obvious that the colour varies from light to dark grey and depends not only on the amount of additive but also on the preparation method. As the data in table 3 reveal, the white colour of the BC matrix has changed significantly; the lightness decreased with increasing additive concentration and the colour difference based on the standard white tile of colorimeter ranged from 31 to 48 and from 50 to 65 for IMP and SOLVO membranes, respectively.

Water vapour sorption capacity of the pure cellulose is around 8.0 [30]. BC is also a good absorbent, and a quite similar value was obtained for the neat BC membrane (table 3, 7.87%). For hybrid BC-based membranes, however, the water sorption decreased with increasing the additive content. Furthermore, due to the more pronounced hydrophobicity, the SOLVO membranes displayed a slightly lower water sorption than the more hydrophilic IMP membranes. In addition, water sorption data were also calculated relative to the amount of cellulose in each membrane (table 3, data in brackets). Interestingly, both membranes with an additive content of 80 absorbed significantly more water

**Table 3.** Lightness, colour difference values and water vapour sorption capacity of BC and BC-based hybrids membranes.

| sample | lightness (*L*) | colour difference[a] ($\Delta E_{ab}$) | water sorption capacity[b] (%) |
|---|---|---|---|
| IMP 20 | 63 ± 2 | 31 ± 2 | 5.66 (7.07) |
| IMP 50 | 50 ± 1 | 45 ± 1 | 3.03 (6.06) |
| IMP 80 | 46 ± 3 | 48 ± 3 | 1.68 (8.40) |
| SOLVO 20 | 44 ± 4 | 50 ± 4 | 4.67 (5.83) |
| SOLVO 50 | 34 ± 6 | 61 ± 6 | 1.73 (3.46) |
| SOLVO 80 | 29 ± 2 | 65 ± 2 | 1.89 (9.45) |
| BC | — | — | 7.87 |

[a]Based on the standard white tile of colorimeter.
[b]Values in parentheses: calculated on the cellulose content of membrane.

**Table 4.** Mechanical and electrical properties of BC and BC-based hybrid membranes.

| sample | thickness (µm) | tensile strength (MPa) | elongation at break (%) | electrical conductivity (S cm$^{-1}$) |
|---|---|---|---|---|
| IMP 20 | 43 | 16.3 ± 1.9 | 4.6 ± 1.4 | $6.1 \times 10^{-5}$ |
| IMP 50 | 63 | 4.6 ± 0.6 | 2.4 ± 0.7 | $2.3 \times 10^{-4}$ |
| IMP 80 | 65 | 1.1 ± 0.5 | 1.7 ± 0.5 | $1.2 \times 10^{-1}$ |
| SOLVO 20 | 38 | 13.0 ± 2.6 | 6.1 ± 1.6 | 3.1 |
| SOLVO 50 | 48 | 0.9 ± 2.6 | 2.6 ± 0.3 | 30.5 |
| SOLVO 80 | 42 | 6.7 ± 1.7 | 3.0 ± 0.2 | 140.1 |
| BC | 37 | 56.8 ± 6.8 | 4.8 ± 2.0 | — |

vapour (8.40 and 9.45%) than the original BC membrane. A possible explanation of this behaviour is that the additive is also accessible to a certain extent of water vapour.

## 3.5. Tensile strength, tear elongation and electrical conductivity measurements

BC membranes exhibit excellent tensile strength (57 MPa; table 4) with a sufficient elasticity (about 5% elongation), due to the strong hydrogen-bonding structure of cellulose molecules. The addition of ZnO–MWCNT additive to BC disrupts this compact and firm structure. For both types of membrane, the tensile strength decreased significantly even at low additive content of 20. However, the structure of as-prepared membranes is still quite stable and resilient, as can be seen in figure 6b.

It is well-known that carbon nanotubes are often used as conductive filler or coating materials [10]. Addition of MWCNT to the BC contributes to the improvement of electrical properties of cellulose/MWCNT composite. The conductivity of the MWCNT-incorporated cellulose was ($1.4 \times 10^{-1}$ S cm$^{-1}$), based on the total cross-sectional area, containing 9.6 w/w% of MWCNT [19]. Similar value ($1.2 \times 10^{-1}$ S cm$^{-1}$) was measured for the IMP 80 membrane (table 4). Comparing the electrical conductivity values of the different membranes, it can be concluded that the BC-based hybrid membranes synthesized by solvothermal method have significantly higher electrical conductivity for each composition than those prepared by impregnation technique. The values of SOLVO membranes increased from 3.1 to 140.1 S cm$^{-1}$ with increasing the additive content. For IMP membranes, however, the values were in a significantly lower range ($6.1 \times 10^{-5}$–$1.2 \times 10^{-1}$ S cm$^{-1}$). Consequently, the electrical conductivity of as-prepared membranes can be tailored by the appropriate synthesis method.

## 4. Conclusion

In this study, a new type of BC-based hybrid membrane (BC–ZnO–MWCNT) was developed and characterized. Results revealed that by choosing the appropriate synthesis method of ZnO–MWCNT

composite additive (such as impregnation or solvothermal), the physical parameters of the as-prepared membranes can be tailored to specific applications. Zeta potential and contact angle data proved that the surface properties of hybrid membranes were modified significantly in a controllable way by the preparation method and concentration of ZnO–MWCNT composite additive. The zeta potential is positive for the SOLVO 80 and SOLVO 90 membranes over the pH range of drinking water. Furthermore, these membranes have the highest surface area value as it was presented in table 2. These facts could make the SOLVO membrane a promising sorbent material candidate towards negatively charged species present in water such as humic acids and viruses via electrostatic interactions. Similarly, the differences in conductivity values of membranes confirmed also the importance of the synthesis method, the SOLVO membranes showed higher conductivity over the IMP membranes. Raman spectroscopy confirmed that the intensity and position of the D- and G- band of MWCNT in the composite were changed in comparison to those in the pristine nanotube. Thus, based on the results of Raman measurements, the formation of a chemical bond between BC fibres and MWCNT-based inorganic additive can be described as an interaction between two nanometric supra-molecular structures. The uniform structure of ZnO–MWCNT additives and the favourable MWCNT–BC matrix interaction led to excellent mechanical properties and flexibility.

The BC-based hybrid membranes decorated with MWCNT–ZnO and prepared by SOLVO method can emerge as versatile materials for future applications in various areas, such as water purification, catalysis or biotechnology.

Authors' Contributions. B.E.M. performed membrane preparation, optimized the inorganic functionalization, and helped with and performed the investigations (e.g. zeta potential measurement). M.U. addressed the analytical chemical problems in the work. E.C. prepared the BC, and organized and helped in determining the mechanical properties. F.K. performed and analysed the XRD investigations. M.L. performed and analysed the FIB-SEM and EDX. L.S.S. performed and analysed the contact angle measurements. M.S. performed and analysed the Raman investigations. Z.N. contributed to conceiving of the study, designing the study, and in its coordination. All authors contributed to the preparation of the manuscript, and all authors accepted the final version of it.

Data accessibility. This article has no additional data.

Competing interests. We declare we have no competing interests.

Funding. This research was supported by the European Union and the Hungarian Government in the framework of the GINOP 2.3.4-15-2016-00004 and GINOP-2.3.2-15-2016-00027, by the BME-Biotechnology FIKP grant of EMMI (BME FIKP-BIO) and by the National Scientific Research Fund of Hungary (OTKA grant no. K131761).

Acknowledgements. We are thankful for the help and the useful comments of the anonymous reviewers.

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
