## [Reviewer comments · Royal Society Open Science]

Review History

RSOS-200592.R0 (Original submission)

Review form: Reviewer 1

Is the manuscript scientifically sound in its present form?

No

Are the interpretations and conclusions justified by the results?

Yes

Is the language acceptable?

Yes

Do you have any ethical concerns with this paper?

No

Have you any concerns about statistical analyses in this paper?

No

Recommendation?

Accept with minor revision (please list in comments)

Comments to the Author(s)

The manuscript by Nemeth et.al. provides that the structural and mechanical properties of BC-MWCNT/ ZnO are tailored by impregnation and solvothermal methods. These two methods are key procedures for tune and decorate the BC-MWCNT/ZnO hybrid membranes. Thus, Author should give the detail procedure of impregnation and solvothermal methods in Materials and Methods section of the manuscript. Even though it cited in the references 22.

The written conclusions is not specific and clear to understand, that which method is provided the finely tuned or excellent morphology of hybrid membranes. Need to rewrite the conclusions before it is considered for the publications.

Also it is difficult to understand and follow the given abbreviations of developed materials, need to change such as

BC-ZnO-MWCNT/Imp instead of... BC-MWCNT/ZnO-IMP (page no 5, line 14)

BC-ZnO-MWCNT/Solv instead of... BC-MWCNT/ZnO-SOLVO (page no 5, line 14)

Because both are derived from different methods and not the part of hybrid membranes.

Overall the work is very important contributions in the field surface modifications of the membrane.

Research article can be consider for the publication after revision of above mentioned comments.

Review form: Reviewer 2**Is the manuscript scientifically sound in its present form?**

Yes

Are the interpretations and conclusions justified by the results?

Yes

Is the language acceptable?

Yes

Do you have any ethical concerns with this paper?

No

Have you any concerns about statistical analyses in this paper?

No

Recommendation?

Accept with minor revision (please list in comments)

Comments to the Author(s)

Minor revision is suggested as below:

1. The amorphous status of ZnO in the composite was verified by X-ray diffraction. It is recommended to obtain selected area electron diffraction of the amorphous ZnO. X-ray diffraction is a mean result of both ZnO NPs and the surrounding materials. TEM can give you an electron diffraction of a much small area.

2. When identifying ZNO crystal structure, the diffracted angles are provided and look like matching well with published data. But it would be very help for the lattice parameters of ZnO crystal can be provided in the manuscript for the convenience of readers' understanding.

3. Raman spectra shows the peaks of MWCNT. They are assigned to D-, G- and G* - bands. Please explain the physical meaning of these bands and provide a reference to support.

4. SEM-Element map in Figures 1d and 1f are not corresponding to the figures 1c and 1f, or 1d and 1e. It is obvious that the shape of the ZnO particle in Figure 1g is very different as that in Figure 1f. Please provide a map that can match SEM micrograph.

5. It is recommended SEM FIB preparing to obtain an XTEM specimen followed by higher magnification TEM investigation in order to provide better proof of MWCNT in terms of lattice fringes showing the layer number of the walls, the inner and outer diameter of the CNT.

it is recommended to accept it on the condition that the above revision request was carefully responded by the authors.

Decision letter (RSOS-200592.R0)

Dear Dr Nemeth:

Title: Development of bacterial cellulose-MWCNT/ZnO hybrid membranes: a study of structural and mechanical properties
Manuscript ID: RSOS-200592

Thank you for submitting the above manuscript to Royal Society Open Science. On behalf of the Editors and the Royal Society of Chemistry, I am pleased to inform you that your manuscript will be accepted for publication in Royal Society Open Science subject to minor revision in accordance with the referee suggestions. Please find the reviewers' comments at the end of this email.

The reviewers and handling editors have recommended publication, but also suggest some minor revisions to your manuscript. Therefore, I invite you to respond to the comments and revise your manuscript.

Because the schedule for publication is very tight, it is a condition of publication that you submit the revised version of your manuscript before 15-May-2020. Please note that the revision deadline will expire at 00.00am on this date. If you do not think you will be able to meet this date please let me know immediately.

Kind regards,
Dr Laura Smith
Publishing Editor, Journals

On behalf of the Subject Editor Professor Anthony Stace and the Associate Editor Dr Dattatray Late.

RSC Associate Editor:
Comments to the Author:
have authors studied the doping concentration dependence studies.

Reviewer comments to Author:
Reviewer: 1

Comments to the Author(s)
The manuscript by Nemeth et.al. provides that the structural and mechanical properties of BC-MWCNT/ ZnO are tailored by impregnation and solvothermal methods. These two methods are key procedures for tune and decorate the BC-MWCNT/ZnO hybrid membranes. Thus, Author

should give the detail procedure of impregnation and solvothermal methods in Materials and Methods section of the manuscript. Even though it cited in the references 22.

The written conclusions is not specific and clear to understand, that which method is provided the finely tuned or excellent morphology of hybrid membranes. Need to rewrite the conclusions before it is considered for the publications.

Also it is difficult to understand and follow the given abbreviations of developed materials, need to change such as

BC-ZnO-MWCNT/Imp instead of... BC-MWCNT/ZnO-IMP (page no 5, line 14)

BC-ZnO-MWCNT/Solv instead of... BC-MWCNT/ZnO-SOLVO (page no 5, line 14)

Because both are derived from different methods and not the part of hybrid membranes.

Overall the work is very important contributions in the field surface modifications of the membrane.

Research article can be consider for the publication after revision of above mentioned comments.

Reviewer: 2

Comments to the Author(s)

Minor revision is suggested as below:

1. The amorphous status of ZnO in the composite was verified by X-ray diffraction. It is recommended to obtain selected area electron diffraction of the amorphous ZnO. X-ray diffraction is a mean result of both ZnO NPs and the surrounding materials. TEM can give you an electron diffraction of a much small area.
2. When identifying ZNO crystal structure, the diffracted angles are provided and look like matching well with published data. But it would be very help for the lattice parameters of ZnO crystal can be provided in the manuscript for the convenience of readers' understanding.
3. Raman spectra shows the peaks of MWCNT. THEY are assigned to D-, G- and G*- bands. Please explain the physical meaning of these bands and provide a reference to support.
4. SEM-Element map in Figures 1d and 1f are not corresponding to the figures 1c and 1f, or 1d and 1e. It is obvious that the shape of the ZNO particle in Figure 1g is very different as that in Figure 1f. Please provide a map that can match SEM micrograph.
5. It is recommended SEM FIB preparing to obtain an XTEM specimen followed by higher magnification TEM investigation in order to provide better proof of MWCNT in terms of lattice fringes showing the layer number of the walls, the inner and outer diameter of the CNT.

it is recommended to accept it on the condition that the above revision request was carefully responded by the authors.

Author's Response to Decision Letter for (RSOS-200592.R0)

See Appendix A.

Decision letter (RSOS-200592.R1)

Dear Dr Nemeth:

Title: Development of bacterial cellulose-ZnO-MWCNT hybrid membranes: a study of structural and mechanical properties
Manuscript ID: RSOS-200592.R1

It is a pleasure to accept your manuscript in its current form for publication in Royal Society Open Science. The chemistry content of Royal Society Open Science is published in collaboration with the Royal Society of Chemistry.

On behalf of the Subject Editor Professor Anthony Stace and the Associate Editor Dr Dattatray Late.

RSC Associate Editor
Comments to the Author:
Authors have revised the manuscript as per referee comments now suitable for publication.

Reviewer(s)' Comments to Author:

Appendix A

University of Miskolc
Institute of Chemistry
H-3515 Miskolc-Egyetemváros
Tel: +36-46-565-111/1380 Fax: +36-46-565-115
E-mail: kemnemet@uni-miskolc.hu

Miskolc, May 14, 2020

Dear Dr. Laura Smith,

I would like to resubmit the enclosed paper RSOS-200592: *Development of bacterial cellulose-ZnO-MWCNT hybrid membranes: a study of structural and mechanical properties* for review and publication in Royal Society Open Science (RSOS). In this manuscript, we report the synthesis of a self-supported and flexible hybrid membrane based on bacterial cellulose (BC) modified with composite additives based on multi-walled carbon nanotube (MWCNT). By choosing the appropriate synthesis method, the size of ZnO, the type of MWCNT/ZnO composite and the physical parameters of the as-prepared BC-based hybrid membranes (e.g. electrical conductivity, surface properties) can be tailored to the specific application. The aim of the study was to work out a controlled process which provides different BC-based hybrid membranes. Furthermore, a comprehensive study about the physical and mechanical properties of BC-based hybrid membranes is presented.

We express our gratitude to the reviewers for the comments on the original manuscript. We considered all of them and revised the manuscript accordingly. We believe that the resubmitted manuscript is appropriate for publication in Royal Society Open Science based on the aims & scope of the journal. This manuscript has not been published and is not under consideration for publication elsewhere. We have no conflicts of interest to disclose.

Sincerely yours,

Zoltán Németh, PhD
Senior Research Fellow

Reply to the comments of Reviewer 1:

Comment 1:

The manuscript by Nemeth et al. provides that the structural and mechanical properties of BC-MWCNT/ ZnO are tailored by impregnation and solvothermal methods. These two methods are key procedures for tune and decorate the BC-MWCNT/ZnO hybrid membranes. Thus, Author should give the detail procedure of impregnation and solvothermal methods in Materials and Methods section of the manuscript. Even though it cited in the references 22.

Reply:

Thank you for the remark. Yes, we agree and a more detailed description of the synthesis procedures has been given. (*page 3, 3.2 Preparation of ZnO-MWCNT composite additives – highlighted in yellow*).

Comment 2:

The written conclusion is not specific and clear to understand, that which method is provided the finely tuned or excellent morphology of hybrid membranes. Need to rewrite the conclusions before it is considered for the publications.

Reply:

Conclusion part (page 6.) has been modified, authors tried to give better explanation about the difference of synthesis routes.

“The Zeta potential is positive for the SOLVO 80 and SOLVO 90 membranes over the pH range of drinking water. Furthermore, these membranes have the highest surface area value as it was presented in Table 2. These facts could make the SOLVO membrane a promising sorbent material candidate towards negatively charged species present in water such as humic acids and viruses via electrostatic interactions. Similarly, the differences in conductivity values of membranes confirmed also the importance of the synthesis method, the SOLVO membranes showed higher conductivity over the IMP membranes. The BC-based hybrid membranes decorated with multi-walled carbon nanotube-zinc oxide and prepared by SOLVO method can emerge as versatile materials for future applications in various areas, such as water purification, catalysis or biotechnology.”

Comment 3:

Also it is difficult to understand and follow the given abbreviations of developed materials, need to change such as BC-ZnO-MWCNT/Imp - instead of - BC-MWCNT/ZnO-IMP (page no 5, line 14) BC-ZnO-MWCNT/Solvo - instead of - BC-MWCNT/ZnO-SOLVO (page no 5, line 14). Because both are derived from different methods and not the part of hybrid membranes.

Reply:

Abbreviations of as-prepared hybrid membranes have been changed throughout the text and the title. Applied abbreviations are BC-ZnO-MWCNT/Imp and BC-ZnO-MWCNT/Solvo.

Reply to the comments of Reviewer 2:

Comment 1:

The amorphous status of ZnO in the composite was verified by X-ray diffraction. It is recommended to obtain selected area electron diffraction of the amorphous ZnO. X-ray diffraction is a mean result of both ZnO NPs and the surrounding materials. TEM can give you an electron diffraction of a much small area.

Reply:

Thank you for the remark. We agree with the reviewer comment and suggestion. Although, in the case of cellulose-based hybrid membranes is a real technical challenge to create a TEM grid and to avoid the membrane burning during investigation. Based on these facts it was not possible to analyse the as-prepared membranes by electron diffraction technique. The morphology of ZnO-MWCNT nanocomposite has been examined widely by TEM technique in the following articles:

Y. Zhang et al. Simple fabrication of free standing ZnO/grapheme/carbon nanotube composite anode for lithium-ion batteries. *Mater. Lett.* 184: 235-238 (2016)

E. Bartfai et al. Synthesis, characterization and photocatalytic efficiency of ZnO/MWCNT nanocomposites prepared under different solvent conditions. *J. Nanosci. Nanotechnol.* 19: 422-428 (2019)

Comment 2:

When identifying ZNO crystal structure, the diffracted angles are provided and look like matching well with published data. But it would be very help for the lattice parameters of ZnO crystal can be provided in the manuscript for the convenience of readers' understanding.

Reply:

Yes, we do agree and lattice parameters have been added to the text (*page 4, 4.2 X-ray and Raman spectroscopy and surface area measurement – highlighted in yellow*).

“The lattice parameters of ZnO were determined during Rietveld refinement, values for the impregnated samples are in average $a=3.251\pm0.001 \text{ \AA}$ and $b=5.210\pm0.002 \text{ \AA}$, while the solvothermal synthesis shows $a=3.250\pm0.001 \text{ \AA}$ and $b=5.207\pm0.001 \text{ \AA}$. The theoretical lattice parameters (COD 9004178) are $a=3.249 \text{ \AA}$ and $b=5.203 \text{ \AA}$, in good agreement with the refined values.”

Comment 3:

Raman spectra show the peaks of MWCNT. They are assigned to D-, G- and G*- bands. Please explain the physical meaning of these bands and provide a reference to support.

Reply:

Thank you for the addition. The D-band is assigned to disorder induced by defects and curvature in the nanotube lattice, the G-band is the in-plane vibration of the C-C bonds, while the G' band is the overtone of D band.

D. Baskaran et al. Noncovalent and nonspecific molecular interactions of polymers with multiwalled carbon nanotubes. *Chem. Mater.* 17; 3389-3397 (2005).

Comment 4:

SEM-Element map in Figures 1d and 1f are not corresponding to the figures 1c and 1f, or 1d and 1e. It is obvious that the shape of the ZNO particle in Figure 1g is very different as that in Figure 1f. Please provide a map that can match SEM micrograph.

Reply:

As suggested, the above-mentioned images (Fig 1d, f and g) have been changed and modified. For better fitting and to improve the quality of the paper Figure 1 has been resized.

Comment 5:

It is recommended SEM FIB preparing to obtain an XTEM specimen followed by higher magnification TEM investigation in order to provide better proof of MWCNT in terms of lattice fringes showing the layer number of the walls, the inner and outer diameter of the CNT.

Reply:

Yes, we agree and the above-mentioned remark, but at this moment the HRTEM is not available at University of Miskolc. We have tried to prepare sample by using FIB-SEM equipment and after that analyse with STEM mode with the same microscope, but the quality of the images was unsatisfactory as can be seen below. Based on this fact and results we do not want to insert these images into the main text (Figure 2.) or to the supplementary part. We are really hope the reviewer agree with our opinion and the Figure 2. shows properly the morphology and structure of samples prepared via SOLVO method.